# Optical Properties and Light-Induced Charge Transfer in Selected Aromatic C60 Fullerene Derivatives and in Their Bulk Heterojunctions with Poly(3-Hexylthiophene)

**DOI:** 10.3390/ma15196908

**Published:** 2022-10-05

**Authors:** Maciej Krajewski, Piotr Piotrowski, Wojciech Mech, Krzysztof P. Korona, Jacek Wojtkiewicz, Marek Pilch, Andrzej Kaim, Aneta Drabińska, Maria Kamińska

**Affiliations:** 1Faculty of Physics, University of Warsaw, Pasteura 5, 02-093 Warsaw, Poland; 2Faculty of Chemistry, University of Warsaw, Pasteura 1, 02-093 Warsaw, Poland

**Keywords:** organic electronics, fullerene derivatives, LESR

## Abstract

Fullerene derivatives offer great scope for modification of the basic molecule, often called a buckyball. In recent years, they have been the subject of numerous studies, in particular in terms of their applications, including in solar cells. Here, the properties of four recently synthesized fullerene C60 derivatives were examined regarding their optical properties and the efficiency of the charge transfer process, both in fullerene derivatives themselves and in their heterojunctions with poly (3-hexylthiophene). Optical absorption, electron spin resonance (ESR), and time-resolved photoluminescence (TRPL) techniques were applied to study the synthesized molecules. It was shown that the absorption processes in fullerene derivatives are dominated by absorption of the fullerene cage and do not significantly depend on the type of the derivative. It was also found by ESR and TRPL studies that asymmetrical, dipole-like derivatives exhibit stronger light-induced charge transfer properties than their symmetrical counterparts. The observed inhomogeneous broadening of the ESR lines indicated a large disorder of all polymer–fullerene derivative blends. The density functional theory was applied to explain the results of the optical absorption experiments.

## 1. Introduction

Fullerenes are one of the best known molecules and have been used as building blocks for many various derivatives and larger structures with a wide range of applications, primarily in biomedicine [1], redox flow batteries [2], organic transistors [3], and photovoltaic nanostructures [4,5]. Among their many derivatives, the commercially available PC61BM ([6,6]-phenyl-C61-butyric acid methyl ester) and PC71BM (([6,6]-phenyl-C71-butyric acid methyl ester) are the best known. Organic and perovskite photovoltaics takes advantage of good electron conductivity and acceptor properties of fullerenes, enabling an efficient transport of photo-excited electrons. In organic solar cells, an active layer typically contains a blend with percolating regions of donor (polymer) and acceptor materials that together create the so-called bulk heterojunction (BHJ), schematically shown in Figure 1. Fullerene derivatives are a traditional acceptor component of the active layer. Although they currently face strong competition from non-fullerene organic acceptors, exhibiting higher open-circuit voltages and up to 18% efficiency [6,7], they are still a strong alternative to such non-fullerene acceptors, especially in terms of the charge carrier mobility values. Recently, research has been underway on ternary blends containing both fullerene and non-fullerene acceptors [8] to take advantage of the specific benefits of each. Moreover, the use of fullerene derivatives in layered photovoltaic perovskite structures as electron-transporting and hole-blocking layers is becoming very common. Solution coating technologies of organic and perovskite photovoltaics require fullerene derivatives with such a modification of the fullerene cage as to ensure suitable solubility in typical solvents used in layer processing. Moreover, fullerene derivatives should lead to the proper ordering of the layers during the evaporation of the solvent to ensure high mobility of the charge carriers and meeting the required electrical parameters of the fabricated solar cell. Despite the unwavering interest and wide range of applications of fullerene derivatives, the understanding of the effect of a particular fullerene modification on its solubility, ability to organize the solidifying layer, along with charge transfer capability and absorption spectrum, needs more in-depth research.

In this study, four recently synthesized fullerene C_60_ derivatives (named [C60] P1, [C60] P2, [C60] T1, [C60] T2) as such and in blends with conductive polymer poly(3-hexylthiophene) (P3HT) were compared regarding their optical properties and light-induced charge transfer ability and their behavior in exemplary solar cells constructed from these mixtures. PC61BM and its blend with P3HT was used as an appropriate reference.

Together with optical absorption measurements, quantum mechanical calculations were performed using the density functional theory (DFT) and time-dependent density functional theory (TDDFT) methods to gain additional insight into the experimental results [9,10,11]. The energy levels and light absorption probability of the new fullerene derivatives, as well as PC61BM, were determined. For studying the charge transfer process, electron spin resonance (ESR) and time-resolved photoluminescence (TRPL) techniques were applied. As has been shown in previous works, both of these techniques provide crucial information regarding the generation and the separation mechanisms of the charge carriers in polymer–fullerene blends [12]. During the illumination of the blend, the photogeneration of strongly bound (0.3–1.0 eV) exciton (electron–hole pair) takes place. The pair is then spatially separated, resulting in the hole being localized at P3HT and the electron at the fullerene derivative [13,14,15,16,17]. Thanks to the ESR technique, it was possible to compare the mixtures with various fullerene derivatives in terms of their efficiency of formation and then dissociation of excitons. Moreover, the TRPL technique was applied to trace the photo-carriers’ behavior in P3HT–fullerene blends. Radiative recombination and charge separation are two competing processes that lead to light emission and photocurrent generation, respectively. Therefore, the observation of the photoluminescence (PL) dumping in the donor–acceptor mixture compared to the PL decay for the individual components of the blend provides information on the effectiveness of charge separation.

## 2. Materials and Methods

### 2.1. Samples

Four differently functionalized fullerene derivatives (Figure 2) were chosen for the presented studies. Two of them, 61-etyloxycarbonyl-[61-pyren-1-ylomethylo]-1,2-methano [60]fullerene (called [C60]P1 in the paper) and bis-61-carbonyl [pyren-1-ylomethylo]-1,2-methano [60]fullerene ([C60]P2), were functionalized with either one or two pyrene moieties. Analogically, the other two, 61-etyloxycarbonyl-[61-(2-(thiophen-3-ylo)ethylo)]-1,2-methano [60]fullerene ([C60]T1) and bis-61-carbonyl o [2-(thiophen-3-ylo)ethylo]-1,2-methano [60]fullerene ([C60]T2), were functionalized with either one or two thiophene rings. Additionally, the commercially available PC61BM purchased from Ossila was used as reference material. Fullerene derivatives [C60]P1, [C60]P2, [C60]T1, and [C60]T2 were synthesized using the modified Bingel method [18,19,20]. Details regarding the synthesis and characterization of the obtained fullerene derivatives are reported for pyrene derivatives in our previous publication [21], and those regarding thiophene functionalized fullerenes can be found in Appendix A. Two types of samples were prepared for the purpose of ESR research: pure fullerene derivatives as such and their blends with P3HT. Functionalized fullerenes were dissolved in chlorobenzene and stirred at a temperature of 50 °C for 3 h under an air atmosphere. The solution was then poured into a 3 mm diameter quartz ESR tube and kept at an air atmosphere of 50 mbar pressure for two days to let the solvent evaporate. The mass of all the studied samples was approximately 3 mg. In the case of fullerene-derivative–P3HT blends, a slightly different procedure was used. Firstly, the commercial regioregular P3HT (purchased at Sigma-Aldrich) and fullerene derivative in the form of powders were mixed at a 3:2 molar ratio. Then, chlorobenzene was applied to create a 25 mg/mL solution. The mixture was stirred at a temperature of 50 °C for 3 h and subsequently poured into an ESR tube. Afterward, the sample was left for two days in an air atmosphere of 50 mbar pressure to let the solvent evaporate to obtain thin and glossy films on the inside wall of the ESR tube. The same solutions were used for the remaining techniques (UV-Vis absorption, PL, and TRPL). However, in these cases, small droplets of the solution were applied onto a glass substrate (UV-Vis) or copper foil (TRPL) and left to dry at 60 °C to create a thin film.

### 2.2. Methods

UV-Vis absorption spectra were recorded at room temperature by means of a Varian Cary 5000 UV-Vis-NIR spectrophotometer in the range of 300–800 nm at 1 nm resolution. Deuterium and tungsten halogen lamps were used as a light source, and the signal was collected using dual Si diode detectors.

The density functional theory (DFT) and time-dependent density functional theory (TDDFT) calculations were employed using the B3LYP hybrid functional, which is considered a reliable tool for structures with extended conjugated double bonds that provides optimized geometries, corresponding energies, and related properties. The calculations were performed with the implementation of the above methods in the Gaussian 09 package [22]. UV-VIS spectra were calculated by TDDFT, as implemented in the Gaussian package. Computations were performed with the use of the B3LYP functional, in the base 6-31 G(d). The number of excited states was 108. The geometry of all the studied molecules was optimized, so that their energies corresponded to the minimum on the potential energy surface.

Electron spin resonance (ESR) measurements were performed by a Bruker ELEXSYS 580 X-band (f ≈ 9.4 GHz) ESR spectrometer with TE102 resonance cavity coupled with an Oxford continuous flow cryostat. During experiments, the modulation frequency and modulation amplitude were 100 kHz and 1 G, respectively. The measurement routine was as follows. The sample was cooled down to 2 K in darkness to avoid any photo-excitation that might affect the intensity of ESR signals. Then, the ESR spectra at a wide range of microwave powers (0.474 µW–0.474 mW) and temperatures up to 80 K were collected. Afterward, the sample was cooled back again to 2 K, and the ESR spectrum was measured under the illumination of the sample with a white LED. Subsequently, the whole series of measurements for different microwave powers and temperatures was repeated for the illuminated sample. In the case of low signals, averaging from 3 to 5 acquisitions was performed to improve the signal-to-noise ratio. Due to residual dark signals of some of the samples, the light-induced ESR (LESR) spectrum will further refer in this paper to the difference in the intensity of the signal with and without illumination. Since the ESR spectrometer measured the derivative of microwave absorption, the signal intensities discussed later relate to a double integral of the measured signals. The fitting of the spectrum was performed using the pepper function (for frozen solution) of the Matlab toolbox–Easyspin [23]. The best fit of the spectra was usually obtained using a convolution of Gaussian and Lorentzian curves. As mentioned before, all ESR experiments were performed for the same amount (mass) of the samples. The ESR spectra of different samples, which were compared for the purposes of the research, were produced under conditions identical to the measurement temperature, microwave attenuation, and field modulation.

Time-resolved photoluminescence (TRPL) was measured at room temperature using Ti:sapphire laser excitation at the third harmonic frequency (300 nm). The obtained spectra were recorded in the wavelength range of 300–900 nm at 1 nm resolution by a Hamamatsu C5680-24C streak camera. To ensure a low signal-to-noise ratio, the final spectra were averaged over 120 measurements. The single transient duration was limited to 12 ns by the laser frequency equal to 80 MHz.

## 3. Results

### 3.1. Optical Absorption Spectroscopy

The absorbance spectra of the studied fullerene derivatives were measured in order to find possible differences regarding both their energy range and structure. Figure 3a shows the absorbance normalized to the intensity of the 3.7 eV peak of pure fullerene derivatives (including PC61BM) at room temperature in the energy range of 1.75 to 4.0 eV. The spectra cover the entire measurement region and most likely extend further toward higher energies. The energy range above 4 eV was not available for the measurement due to strong absorption in a glass sample substrate. The absorption spectrum of unmodified C_60_ fullerene molecule measured by other researchers [24] starts with a very weak band (almost invisible in the case of the measured fullerene derivatives), which was identified as a dipole-forbidden transition at a HOMO-LUMO gap of 1.85 eV (S_0_→S_1_ transition). It is followed by the first strong band centered at about 2.7 eV and related to a dipole-allowed transition to the higher excited state (S_0_→S_2_). The third, even stronger peak at 3.6 eV was also ascribed to a dipole-allowed transition, from the lower level in the valence band to the S_1_ level. All the studied fullerene derivatives had similar absorption bands to the ones discussed above—however, with some exceptions. An evident difference was observed for the [C60]P2 absorption, where an additional strong line appeared at about 3.4 eV energy. This line was also present in the [C60]P1 absorption spectrum but as a weaker structure superimposed on the 3.6 eV transition. The line at 3.4 eV may be related to the internal excitation within the pyrene molecule, since its intensity grows with the number of pyrenes attached to the fullerene cage, and it has been found previously that the pyrene monomer exhibits three sharp absorption peaks in the range of 310–340 nm [25], which corresponds well to the energy region of the observed extra line. Further support for such identification comes from the DFT calculations presented in Figure 4 and discussed later. On the other hand, the absorption related to the excitation within the thiophene ring/rings present in the [C60]T1 and [C60]T2 derivatives can be expected in the ultraviolet range, outside of the measurement range [26]. Thus, the results of absorption measurements show only a slight influence of the chemical functionalization of the synthesized fullerenes on the observed spectra, proving the dominance of excitations within the fullerene cage.

Since fullerene absorption occurs in the area of higher energies than the maximum of the solar spectrum, they are not an absorber in the solar cells; however, they can use the high-energy part of the solar spectrum and enhance the operation of the cells. Among the studied fullerenes, it can be noticed that the absorption capability shifts toward lower energies for fullerenes with pyrene moiety, and the intensity of this absorption increases with the number of pyrenes. For this reason, these fullerene derivatives have the potential to be a better complement to solar cell absorbers than the commonly used PCBM and more efficiently enhance solar energy harvesting by supporting the P3HT donor in the process of carrier photogeneration.

In Figure 3b, the fullerene-derivative–P3HT blends and pure P3HT absorption spectra are shown. As seen, P3HT absorption extended in the energy range of 2.0–2.8 eV, with a maximum at 2.4 eV. The peak at 2.4 eV was present in the spectra of all blends. Interestingly, the intensity of this peak in relation to the fullerene peak at 3.6 eV strongly varied depending on the kind of fullerene derivative present in the blend, despite the preparation of all blends at the same nominal ratio of P3HT to fullerene derivative. This behavior may result from some differences in the solubility of the fullerene derivatives (Appendix A).

### 3.2. Density Functional Theory Calculations of Optical Transitions

For optimized geometries, one-electron energy levels, and among them, HOMO and LUMO energies were calculated for all studied fullerene derivatives, including PC61BM. The results along with the corresponding models of HOMO are presented in Table 1. The already published data for the HOMO/LUMO energy levels of PC61BM using the same basis set are consistent with ours within 0.1–0.2 eV [27,28,29].

Due to the size of the studied molecules, to optimize their geometry and calculate the corresponding energies, we used a relatively small double-zeta basis sets 6-31G (d). To compare whether the calculation using larger triple-zeta basis sets can qualitatively change the conclusions resulting from HOMO and LUMO energy estimates, single-point calculations for the optimized geometries were performed using triple-zeta basis sets, i.e., 6-311G (d, p). The energy values obtained in this way for HOMO and LUMO are included in Appendix A. It can be seen that, in general, there is no change in the sequence of levels for all fullerene derivatives, and the energies of the levels are shifted downward by about the same, −0.4 eV, energy. Therefore, the following discussion and conclusions also remain valid in the larger basis sets.

It is seen from Table 1 that the chemical modifications of fullerenes by pyrene substituent ([C60]P1 and [C60]P2 derivatives) resulted in significant changes in the energy of HOMO levels. For the rest of the fullerene derivatives, the HOMO level was practically at the same position as for PC61BM. Analysis of the corresponding orbitals revealed that for the pyrene functionalized fullerenes, the HOMO respective orbital was located on the pyrene moieties, whereas for the other studied molecules, it occupied the fullerene cage. The LUMO energies were similar for all fullerene derivatives, and all the LUMO (responsible for accepting electrons in solar cells) were located on the fullerene cage. One can expect optically forbidden transitions between the orbitals localized at the pyrene moiety and the ones localized at the fullerene cage, so the energies of lower levels (below the HOMO level) for the [C60]P1 and [C60]P2 derivatives are also included in Table 1. In the table, the energies of HOMO levels down to the level with the orbital located at the fullerene cage are shown. The highest occupied molecular orbitals located at the fullerene cage were determined as HOMO-1 for [C60]P1 and HOMO-2 for [C60]P2.

The energy positions of these latter levels are very close to the HOMO level of PC61BM as well as fullerenes modified with thiophene rings. The LUMO-HOMO energy gaps of the studied molecules differed by up to 0.21 eV (Table 1). Such a difference does not correspond to the measured absorption spectra, which, according to Figure 3a, were very similar and in the same energy range. To explain this discrepancy, the intensities of optical transitions were calculated in the energy range of 1.5–4.5 eV. The calculated spectra for all the studied fullerene derivatives along with the spectrum for the unmodified C_60_ fullerene are presented in Figure 4. As can be seen, the spectra were dominated by several peaks around 4 eV; those peaks originated from the splitting of a single line of C_60_ absorption observed at 4.0 eV. This line most probably corresponds to the mentioned dipole-allowed transition from the lower level in the valence band to the S1 level (observed in the optical absorption experiment at about 3.6 eV). Noteworthy are the additional structures appearing in the calculated absorption spectra of fullerene derivatives with pyrene moieties, [C60]P1 and [C60]P2, at about 3.6 eV energy, and not visible for the other fullerene derivatives. They are especially strong for [C60]P2, with two pyrene moieties, but are also seen for the calculated absorption spectrum of the [C60]P1 derivative and can therefore be linked to optical intra-pyrene excitations [25]. We relate this theoretical structure to the one observed in the experimental absorption spectra for [C60]P1 and [C60]P2 at 3.4 eV already discussed in the experimental part (Section 2). The calculated absorption threshold and the spectrum at lower energies were much weaker than the structure at about 4 eV for all fullerene derivatives; however, the positions of the lines and their intensities in the 1.9 to 3.5 eV energy range were very similar for all of them. Therefore, it is possible to conclude that, in this energy range, transitions within the fullerene cage were dominant and made the spectra for all the fullerene derivatives very alike. Transitions from HOMO localized at the pyrene moieties for [C60]P1 and [C60]P2 to LUMO localized at the fullerene cage were not visible in the calculated spectrum. The reason for this absence is the weak overlap of HOMO and LUMO wave functions, making the optical excitation very unlikely. Interestingly, optical transitions from orbitals with energy levels below HOMO:HOMO-1 for [C60]P1 and HOMO-2 for [C60]P2 were visible in the spectrum. These orbitals are the highest occupied orbitals localized at the C_60_ fullerene cage and play the HOMO role in terms of optical absorption. As a result, the effective energy gap of fullerene derivatives functionalized with the pyrene is very close to the energy gap of the other functionalized fullerenes. As can be seen from Table 1 (LUMO–HOMO energy differences marked in bold), such effective energy gaps differ only by 0.02 eV.

The calculated transitions for all fullerene derivatives, as shown in Figure 4, seemingly start at a lower energy than the LUMO–HOMO distance. This is related to the DFT calculation method that takes into account the reconfiguration of the charge density distribution after electron excitation (e.g., from HOMO to LUMO). Therefore, the energies corresponding to such reconfigured excited states may differ from those determined for the original shape of the orbital. Additionally, in general, the calculated absorption structures are shifted to higher energies compared to the measured spectra. This, however, is typical for the DFT/B3LYP/6-31G(d) method, mainly due to the overestimation of LUMO energy [28,30]. It is worth noting that only the slightly different values of the HOMO and LUMO levels of the studied fullerene derivatives reveal that all the fullerene derivatives can work with the same donor materials in solar devices.

### 3.3. Electron Spin Resonance

#### 3.3.1. Signal Analysis

As mentioned above, two kinds of samples were measured using the ESR technique: pure fullerene derivatives and their blends with P3HT (Figure 5). In darkness, most fullerene derivatives did not show any significant ESR signal (data in Appendix A). Some of them exhibited a persistent line of g = 2.0022, which has been attributed earlier to either residual oxygen ions [31] or other intrinsic defects, which may give rise to unpaired spins localized at deep traps [32,33]. Nevertheless, these signals were much weaker than those obtained under illumination and were subtracted during the analysis. As can be seen in Figure 5a, fullerene derivatives exhibited the LESR signal with two lines of various intensities. The line of g = 2.0000 can be related to an electron localized at the fullerene cage due to its similarity to the line identified as such and observed in polymer–fullerene blends [34]. The other line of g = 2.0022 has been usually assigned to a hole localized at the polymer chain in polymer–fullerene blends; however, in this case, the signal might arise from a hole localized at the fullerene functional group. Symmetrical molecules ([C60]P2, [C60]T2) demonstrated a much weaker signal at g = 2.0000 compared to what was observed for asymmetrical ones ([C60]P1, [C60]T1, and also PC61BM). Regarding the line corresponding to the holes, it was weaker for molecules with a pyrene moiety ([C60]P1, [C60]P2), so they seem to be less likely to accumulate holes as compared to those with the thiophene ring ([C60]T1, [C60]T2). The reference sample, PC61BM, exhibited a very similar signal to [C60]T1, yet much more prominent.

Fullerene derivative blends with P3HT exhibited substantially different ESR spectra than pure fullerene derivatives. In this case, blends with asymmetrical fullerene molecule ([C60]T1:P3HT, [C60]P1:P3HT) gave rise to a strong single line at g = 2.0025 under dark conditions. This signal may be attributed to intrinsic defects inside the molecules [33]. For symmetrical molecule blends ([C60]T2:P3HT, [C60]P2:P3HT), this dark signal was of much lower intensity. Under illumination, two lines appeared in all blends, with g factors as those observed for pure fullerene derivatives. The main noteworthy difference between pure fullerene derivative spectra and those for blends with P3HT was significant signal intensity growth in the case of the blends. Interestingly, for the asymmetric molecule blends, the line intensities of the hole and the electron components gave stronger LESR signals than for the symmetric molecule blends, with a dominant hole component for [C60]T1:P3HT, [C60]P1:P3HT and almost equal intensities of both signals for PC61BM:P3HT. The blends with symmetrical fullerene molecules ([C60]P2:P3HT, [C60]T2:P3HT), on the other hand, exhibited generally lower but similar intensities of the electron and hole components. Thus, the following correlation could be observed: the stronger the LESR signal detected for the pure fullerene derivative, the stronger the LESR signal for its blend with P3HT. To make a more quantitative comparison of the separated electrons and holes in the studied blends, the LESR signals were fitted using the procedure described in Section 2.2.

For accurate fitting, an anisotropic g-tensor, which indicates the existence of some preferential order inside the blends, causing differences in magnetic susceptibility along the respective directions, was necessary. All components of the g-tensor along with the average g-factor attributed to either the hole or the electron signal, (*g_iso_*) are listed in Table 2. The components of the g-tensor for a photo-generated hole in all P3HT blends with different fullerene derivatives are consistent within the experimental uncertainty of ±0.0005 and also agree with the literature values for the PC61BM:P3HT blend [13]. This indicates that a photo-generated hole localized at the polymer backbone does not interact noticeably with electrons localized at the fullerene cage. The absence of such interactions means that photo-induced electron–hole pairs inside these blends are quickly separated [15]. Indeed, fast charge separation has been observed in a PC61BM:P3HT blend, with a reported time of about 40 fs [35,36]. The g-tensor components of the photo-generated electron localized at the fullerene cage in all blends with P3HT were very similar to those in pure fullerene derivatives (data in Appendix A), and all of them were well below the free electron g-factor (g = 2.0023). Such negative deviation most probably results from the spin–orbit coupling with the unoccupied π-orbital, while positive deviation has been considered to arise from the spin–orbit coupling with the occupied orbitals [37,38]. The LESR signals of electrons at the fullerene cage showed only slight differences between the various molecule blends, indicating a rather weak influence of the different fullerene functional groups. However, g-tensors exhibited a different degree of anisotropy. In the case of asymmetric molecules, the highest and uniaxial anisotropy was observed for PC61BM, a little smaller and still uniaxial for [C60]T1, followed by [C60]P1 with no axial symmetry. Interestingly, in the case of symmetric molecules, the g-tensor for [C60]P2 showed very small anisotropy, whereas for [C60]T2, it was completely isotropic. The g-factor anisotropy may indicate a certain ordering of the fullerene molecules; therefore, the obtained results would indicate the best order for the PC61BM:P3HT blend.

#### 3.3.2. LESR Signal Saturation

In order to study the properties of the fullerene derivative with P3HT blends, the LESR spectra as a function of microwave power were measured for the temperature range 5 K–60 K. In Figure 6, for clarity, only the data for the three selected temperatures are shown. As can be seen, the LESR signals tended to saturate, and as the temperature increased, a shift of the microwave power saturation point toward higher power values was observed. At low temperatures, the LESR signals were quickly saturated both for the electrons and the holes, indicating longer relaxation times than for higher temperatures. At 59 K, the electron and hole signals started to saturate at the same microwave power, which was about 0.05 mW.
(1)$$I=A\sqrt{P}{\left[1+\left(2^{\frac{1}{\varepsilon }}-1\right)\frac{P}{P_{\frac{1}{2}}}\right]}^{-\varepsilon }$$
where *A* is the scaling factor, *p* is the microwave power, *P*_1/2_ is the microwave power at which the first derivative amplitude is reduced to half of its unsaturated value, and ε is the homogeneity factor. When the saturation originates from inhomogeneous broadening, *ε* = 0.5, while for homogeneous broadening, *ε* = 1.5.

A saturation of the ESR signal is often observed for higher microwave powers due to the finite relaxation time of the excited electrons with spin reversal by microwaves [39]. Generally, spin may relax its polarization but also coherence through the so-called spin–lattice or spin–spin interaction mechanisms. Spin–lattice relaxation involves the exchange of energy with lattice vibrations, causes longitudinal (polarization) relaxation of the spin, and its characteristic time is typically denoted as *T*_1_. On the other hand, spin–spin interaction (with energy conservation) is effective in destroying coherence, and it is characterized by *T_2_* time. The saturation mechanism includes two extreme cases, called homogeneous and inhomogeneous broadening. The source of homogenous broadening is fluctuating fields, originating mostly from the electron and nuclear spin flips, occurring for an ensemble of spin systems experiencing the same time-averaged local fields. Such situation is typical for paramagnetic centers in single crystals. The width of a homogenous line is given by 2/*T_m_*, and *T_m_* means the phase relaxation time, and it is often identical to *T*_2_. On the other hand, in an inhomogeneous case, the ESR signal is a superposition of lines from spin ensembles with different Larmor frequencies and typically has a Gaussian shape with a width determined by inhomogeneity. The important sources of inhomogeneous broadening are different orientations of the grains, lack of a long-range order, or changes in the local environment of the spin ensembles in disordered solids (powder, glass, amorphous materials, irradiated or strained crystals), and the inhomogeneous broadening originates from the resulting anisotropy or slight changes of the interaction tensors. Additionally, hyperfine interactions cause inhomogeneous broadening. Whereas the ESR signal intensity is proportional to the square root of microwave power for a low microwave power range in both cases [39], its intensity reaches a maximum and then decreases with increasing microwave power in the case of homogenous broadening. In turn, for inhomogeneous broadening, the ESR signal intensity increases and remains almost at a constant value with increasing microwave power. The following formula describes the changes in ESR signal intensity as a function of microwave power [40,41].

To examine the source of the LESR line broadening, we fitted the experimental intensity of the LESR signal (double integral of the collected signal) as a function of microwave power using Formula (1) with *ε* as a fitting parameter. The fitted dashed curves are shown in Figure 6, and they reproduce the observed dependencies very well. The obtained values of the *ε* fitting parameter are presented in Appendix A. As can be seen, for most of the curves, the value of the *ε* parameter is close to 0.5 with an uncertainty of 0.1. Such estimated error of fitting is mostly a result of experimental data processing—primarily the partial overlapping of LESR signals from the electron and the hole, low intensity of the signals in the low microwave powers range, as well as background subtraction. The obtained value of the *ε* parameter indicates the inhomogeneous origin of the LESR signal broadening. This is due to a significant disorder of the polymer–fullerene blend structure resulting in dispersion of the g-tensor values. A slightly higher value of the *ε* parameter, reaching 0.7 in a few cases, but still far from 1.5 (characteristic value for homogenous broadening), indicates the dominant contribution to the LESR line width from the disorder of the blend structures, causing local changes in the g-tensor.

The inhomogeneous origin of the LESR signal broadening means that the *T*_2_ relaxation time cannot be derived from the LESR line full width at half maximum (FWHM). The FWHM size is rather a measure of the degree of variation in the local environments of photo-excited carriers [42]. The smallest line widths for both the polymer and the fullerene derivatives were recorded for the mixture of P3HT and PC61BM. This means the smallest degree of differentiation around the corresponding paramagnetic centers and suggests the best ordering of this mixture regarding both of its components. However, generally, the ESR studies indicate a high degree of heterogeneity in the vicinity of the paramagnetic centers, photo-generated electrons, and holes, which proves, in line with the intuition, a high degree of disorder of the polymer–fullerene blends. In agreement with our conclusion about inhomogeneous broadening in polymer–fullerene blends, the authors of Ref. [43] have attributed the wide distribution in longitudinal relaxation times (*T*_1_) to the strong heterogeneity of the blend and the presence of energetically distributed traps.

### 3.4. Photoluminescence

The kinetics of photoluminescence spectra, TRPL, were measured at room temperature for pure P3HT, pure fullerene derivatives, and blends of P3HT with the fullerene derivatives. Such studies allowed us to observe the influence of the blend components on exciton recombination. The selected TRPL results (a complete set of spectra related to the [C60]P1 fullerene derivative) are shown in Figure 7, while the TRPL spectra for all compounds studied are presented in the Appendix A. Radiative recombination of charge carriers originating from excitonic excitation is mostly governed by first-order kinetics; therefore, the concentration of excitons (*n*) should decay exponentially with time.
(2)$$n\left(t\right)=Aexp\left(-\frac{t}{\tau }\right)$$
where *A* is a proportionality constant, and *τ* is the exciton decay time. The recombination rate of the studied materials is dominated by an exponential decay, clearly visible in the long time range (>0.5 ns) (in Figure 7, the contours are then approximately equidistant for a fixed energy). P3HT has quite strong luminescence with a maximum at about 1.8–1.9 eV and a decay time of about 0.2 ns. The [C60]P1 fullerene derivative has about 10 times weaker luminescence intensity. It radiates briefly (0.05 ns) in a broad band of 1.6–2.1 eV and then continues to emit a longer-lasting (ca. 0.5 ns decay time) signal at about 1.7 eV. Figure 8 shows in-sequence time-integrated photoluminescence (PL) spectra of the studied fullerene derivatives in comparison with the spectra of P3HT and the relevant fullerene-derivative–P3HT blend. All fullerene derivatives exhibited a PL peak at 1.7 eV. The P3HT spectrum was about an order of magnitude stronger than those of fullerene derivatives when comparing their maxima (note the logarithmic scale of PL intensity) and consisted of two broad lines at approximately 1.8 eV and 1.9 eV. In the PL spectrum of P3HT blends, it was possible to distinguish structures originating from P3HT and the respective fullerene derivative. In general, all spectra of the blends were attenuated compared to the spectra of the pure components. This was particularly evident for the P3HT contribution, which decayed almost equally strongly in all blends, and proved a fast transfer of electrons from P3HT to the respective fullerene derivative before these electrons could recombine radiatively within the exciton pairs in P3HT. Thus, the blends with all the fabricated fullerene derivatives showed similarly efficient transport of photo-generated electrons from P3HT to fullerenes through the LUMO states. What is worth noticing is that the blend of PC61BM:P3HT exhibited strong and fastest decay of the whole PL spectrum, not only in the part related to recombination within P3HT but also in the part related to PC61BM. The analogous components of the blend spectra with fullerene derivatives fabricated in this work showed generally weaker quenching of the PL spectrum related to fullerene recombination. Such observation can be explained by the weaker transfer of holes from the new fullerene derivatives to P3HT compared to the hole transfer from PC61BM. This may indicate an existence of a barrier between the HOMO levels of the new fullerene derivatives and P3HT, in contrast to the situation in the PC61BM: P3HT blend in which this transport is smooth.

A rather unusual TRPL spectrum was recorded for the [C60]P2: P3HT blend; it contained a significantly greater P3HT contribution than in the other cases. This may be due to improper phase separation inside the photosensitive layer, mainly due to the low solubility of the fullerene derivative [C60] P2, which excludes part of the derivative [C60] P2 from forming a relatively homogeneous blend phase. This is an issue, which was already discussed above in the context of the absorption spectra for the blends. The solubility table in the Appendix A confirms these assumptions.

To analyze qualitatively the time-resolved luminescence of pure fullerene derivatives and compare it with their luminescence decay in the respective blends with P3HT, all TRPL spectra were integrated in the energy range corresponding to the spectral range of fullerene luminescence (1.54–1.8 eV). The results are shown in Figure 9**.** In other words, Figure 9 presents luminescence decays over time of the studied fullerene derivatives, both pure and in blends with P3HT. Data for [C60]P2 are not presented here due to the solubility problems of this fullerene derivative, whereby the luminescence of the blend was dominated by the P3HT luminescence (see Figure 8). It can be noticed that the decay of luminescence in blends was faster than that of pure fullerene in all cases except [C60]T2, for which both decays were the same. For pure fullerene derivatives, the TRPL curves were exponential in the long time range (>0.5 ns), which is observed as a straight line in Figure 9, with characteristic times of about 0.4 ns (estimated from fits created in this time range, as shown in Figure 9). This time region can be assigned to radiative recombination with recombination times typical for organic materials. For a short time range, up to about 0.3 ns, the photoluminescence decay occurred faster and was a result of the existence of defect centers in fullerene derivatives trapping the carriers or causing their non-radiative recombination and competing with radiation processes. The TRPL curves for the fullerene derivatives in blends had a similar exponential decay in the long time range, only shifting down in parallel to the curves for pure fullerene derivatives, with similar characteristic times of about 0.4 ns. As mentioned, this long time region can be assigned to the typical radiative recombination. The main difference between the decays of the curves for pure fullerene derivatives and those derivatives in blends occurred in the area of short times, up to about 0.4 ns. In this range, the decay of the photoluminescence occurred faster in the case of blends, which had to be a result of the photo-excited hole transfer from fullerene to P3HT. As can be seen, the hole transfer to P3HT occurred with greater or lesser efficiency in different blends, except for [C60]T2, in which this transfer was imperceptible when it came to TRPL curves. On the other hand, from Figure 9, it is obvious that the hole transfer was most effective in the case of the blend with PC61BM. These results correspond to LESR measurements where the most intensive signals were from the P3HT: PC61BM blend, and symmetrical molecules exhibited the lowest signal intensities, proving a weaker separation of the charges. To quantitatively describe the contribution of various processes to the luminescence decay and to estimate the characteristic times associated with them, we assumed that the decay rate of the exciton in a blend γ, reciprocal to its decay time $\gamma =1/\tau_{blend}$, is a sum of radiative recombination ($\gamma_{RR}$) and carrier escape ($\gamma_{E}$) rates: $\gamma =\;\gamma_{RR}+\;\gamma_{E}$. In the above, we approximated the radiative decay rate by the decay rate for pure fullerene derivatives (and similar rate for fullerenes in blends) γRR≈1τpure in the long time range as about 0.4 ns. The blend decay rate should therefore be equal to
(3)1τblend =1τpure+ γE

$\tau_{blend}$ was estimated from the 0–0.4 ns range of the TRPL transients by the fitting curve
(4)$$I\left(t\right)=I_{0}e^{-\frac{t}{\tau }}+I_{inf}$$
where *I_inf_* represents an emission of lifetime so long, that it is infinite in this time range. The obtained curves are plotted in Figure 9 in the respective fitting ranges.

For all the studied samples, it was about 40–80 ps. The calculated $\gamma_{E}$ from Formula (4) exhibited the highest value for PC61BM (22 ns^−^^1^), followed by the [C60]P1, [C60]T2, and [C60]T1 fullerenes with values of about 8–17 ns^−^^1^ (detailed results are presented in Appendix A). This result quantitatively shows that the most efficient hole transfer takes place in the PC61BM:P3HT reference sample.

### 3.5. Preliminary Solar Cells Construction with the Newly Synthesized Fullerenes

Organic solar cells with an active layer in the form of a bulk heterojunction with the P3HT polymer were fabricated from the studied fullerene derivatives. The details regarding the manufacturing process and electrical parameters of the cells are provided in the Appendix A, and they performed significantly worse than the reference cell with the PC61BM acceptor. Among the new fullerene derivatives, the cells with fullerenes functionalized with one aromatic substituent, i.e., [C60]P1 and [C60]T1, were better. However, they were still characterized by substantially lower short circuit current density (J_SC_) and open circuit voltage (V_OC_), and as a consequence, lower power conversion efficiency (PCE) than the reference cell with PC61BM. The cells with symmetric fullerene derivatives, functionalized with two aromatic substituents, [C60]T2 and [C60]P2, underperformed even more. Some improvement in cell parameters could probably be achieved through dedicated optimization for newly functionalized fullerenes; yet, this would require separate, in-depth research, which was not our goal at the moment. Secondly, the differences in the solubility of the synthesized fullerene derivatives certainly influenced the final structure of the active layer, and thus, its charge transport properties. Insufficient solubility was observed mainly for [C60] P2, where the PL signal was dominated by P3HT. This may be related to the sediments observed in the active layer. The investigated fullerene derivatives seem to have problems in forming a proper volumetric heterojunction due to their tendency to form thick, impermeable clusters or their inability to easily blend with the conductive P3HT polymer, the crucial factors influencing the final efficiency of the blend [44,45,46].

The results of the presented research suggest that asymmetrical, dipole-like structures are more likely to create a desired phase-separated active layer. Additionally, it is important to note that the differences in PCE values were caused not only by the difference in solubility. As shown by the presented research, in blends with the synthesized fullerene derivatives, there was a problem in the transport of holes to P3HT, especially serious for symmetric fullerenes.

## 4. Conclusions

Four aromatic fullerene C_60_ derivatives were synthesized. They were studied to determine the optical properties and the efficiency of charge separation in blends with P3HT. The research showed that the applied chemical functionalization of the fullerene cage leads to only slight changes in the absorption spectra, since the absorption was dominated by excitations within the fullerene cage. Some additional absorption present in the visible range for fullerene derivatives with pyrene moieties was related to the internal excitations within the pyrene molecules. Its intensity increased with the number of pyrene substituents, which functionalized fullerene. Such excitations seem to be a chance for a modification of fullerene derivative absorption, so that the fullerenes can support the polymer as a sunlight absorber.

Light-induced ESR studies revealed two prominent, overlapping lines—one from a positive charge localized at the P3HT polymer and the other from a negative charge localized at the fullerene cage for all the studied blends. The reference blend with PC61BM showed the most intense LESR signal and PL quenching, which was a sign of efficient charge transfer. PC61BM with its good performance was followed by asymmetrical [C60]P1 and [C60]T1 derivatives, whereas the symmetrical [C60]P2 and [C60]T2 demonstrated a much weaker charge separation in the blends. An interesting result of the research is therefore the finding of stronger charge separation properties of asymmetrical derivatives in blends with P3HT compared to the symmetrical ones. This suggests that the interaction and mixing of materials are better in the case of molecules with a dipole moment. Detailed analysis of the LESR spectra saturation demonstrated inhomogeneity of the environments of paramagnetic centers in blends. However, the PC61BM: P3HT reference blend exhibited the lowest degree of heterogeneity for both the photo-excited electron and the hole surroundings.

The above results were consistent with the performance of the preliminary solar cells. The fullerenes functionalized with one aromatic substituent performed better in the active layer than their symmetrical counterparts bearing two aromatic moieties. Thus, in the case of asymmetric fullerene derivatives, their interfaces with P3HT and the overall arrangement of the blends allowed for better charge transfer. An interesting observation is also the inefficiency of the hole transfer from the synthesized fullerenes to P3HT, which was most severe for the symmetric fullerene derivatives.

## Figures and Tables

**Figure 1 materials-15-06908-f001:**
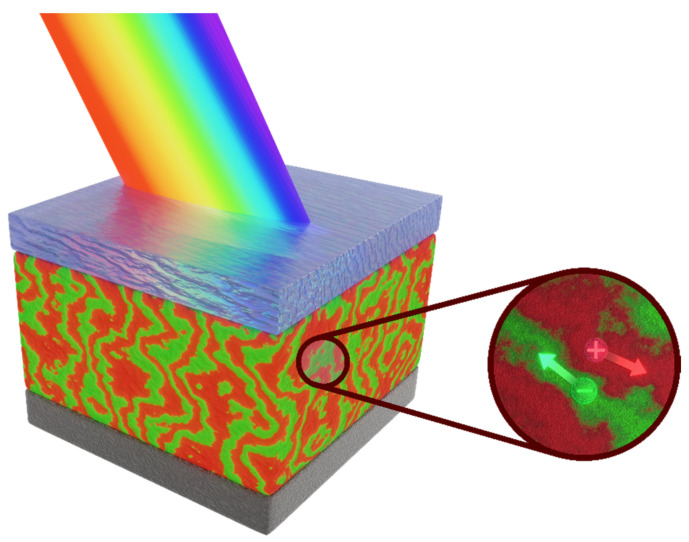
Schematic electron and hole separation in bulk heterojunction photogenerated after illumination of the blend. The red and green arrows on the right scheme show the transport route for photogenerated hole and electron, respectively.

**Figure 2 materials-15-06908-f002:**
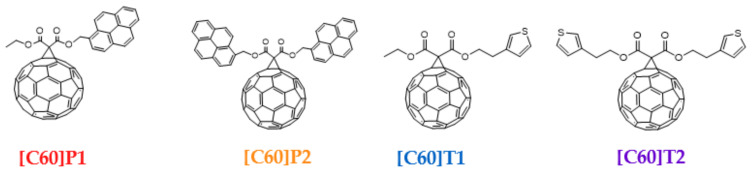
Chemical structure of the studied fullerene derivatives.

**Figure 3 materials-15-06908-f003:**
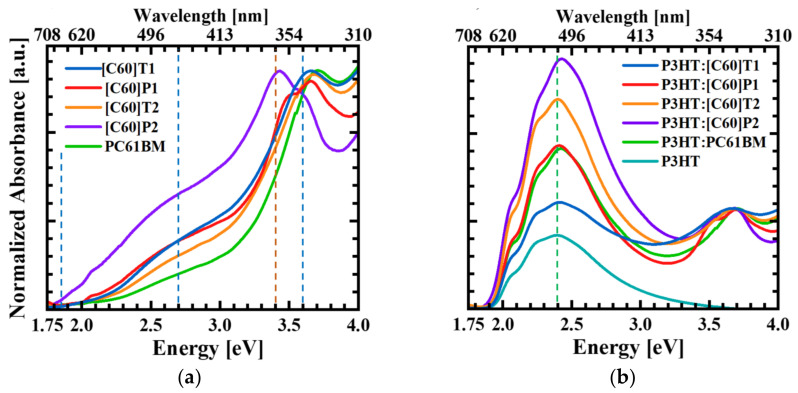
Absorption spectra of (**a**) fullerene derivatives and (**b**) their blends with P3HT. The dashed lines mark the respective transitions: the intra-pyrene transition is marked with the orange line, fullerene transitions are marked with the dark blue lines and P3HT maximum absorption with the green line.

**Figure 4 materials-15-06908-f004:**
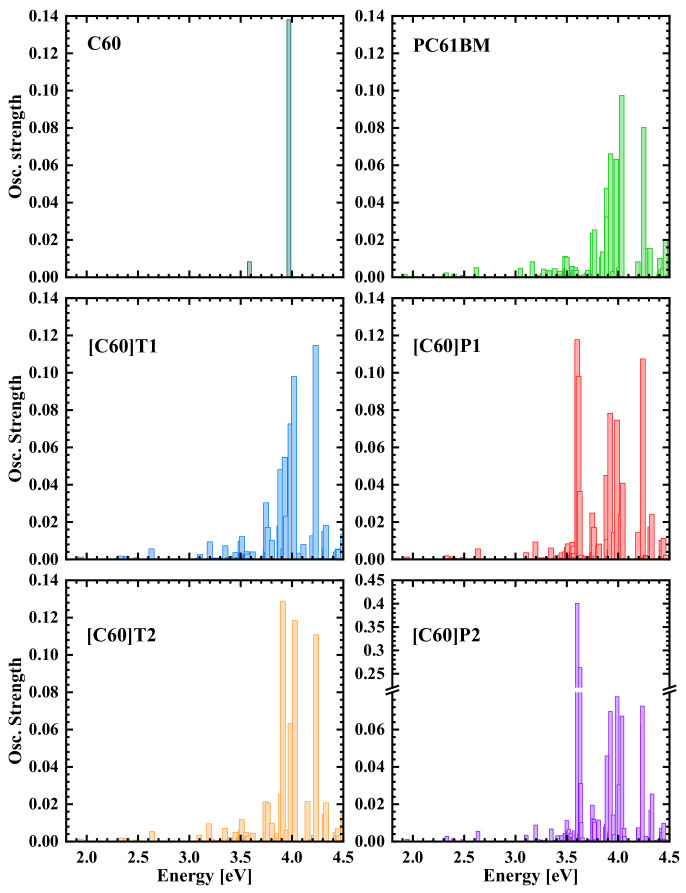
Calculated oscillatory strength of optical transitions for the studied fullerene derivatives and C60 molecule. Note the broken scale in the ordinate axis for the [C60]P2 spectrum (strong intra-pyrene transitions).

**Figure 5 materials-15-06908-f005:**
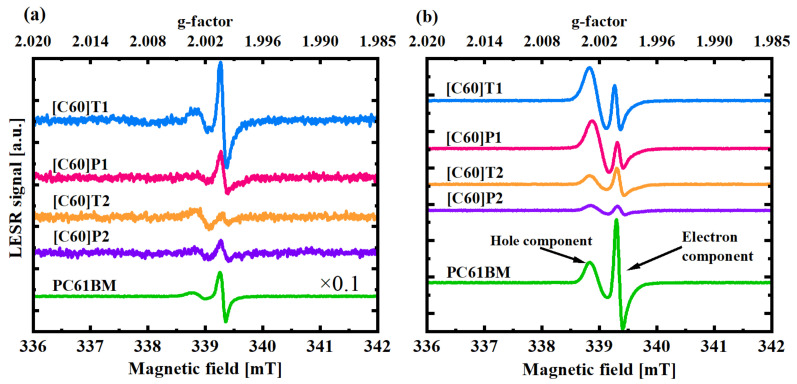
LESR signal at 30 K and for 0.00474 mW of microwave power of (**a**) differently functionalized fullerene derivatives; note that the reference signal of the PC61BM fullerene was divided by 10 due to its high intensity, (**b**) fullerene derivative blends with P3HT. The signals are vertically shifted for better clarity with the scale maintained.

**Figure 6 materials-15-06908-f006:**
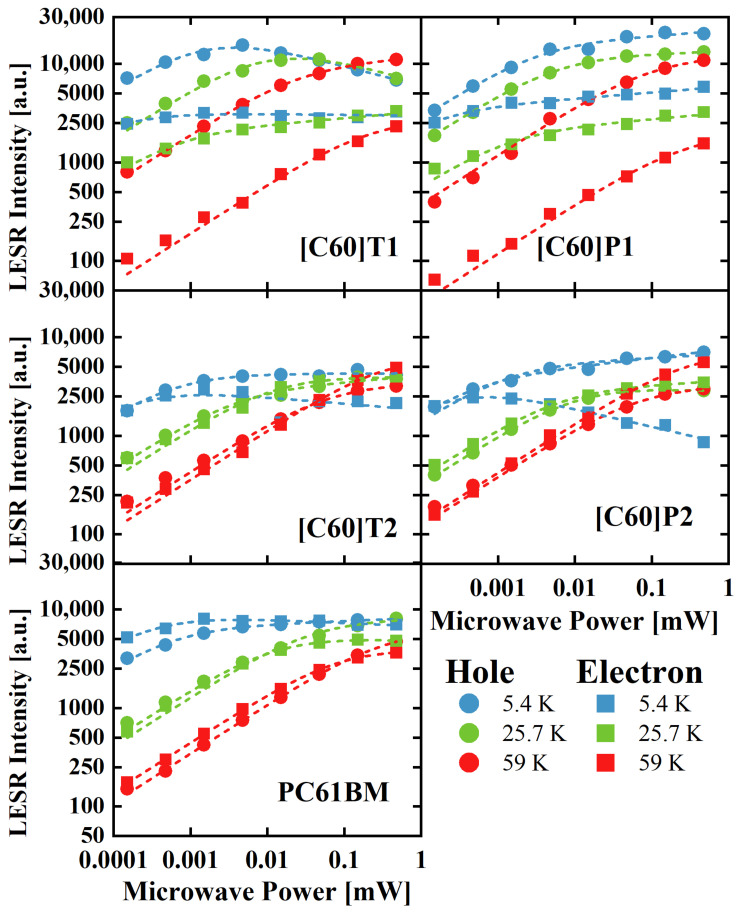
LESR signal intensities as a function of microwave power shown in log–log scale. The indicated fullerene derivatives were measured in blends with P3HT at different temperatures (the results for three selected temperatures are presented). The dashed lines were fitted with the experimental points using Formula (1).

**Figure 7 materials-15-06908-f007:**
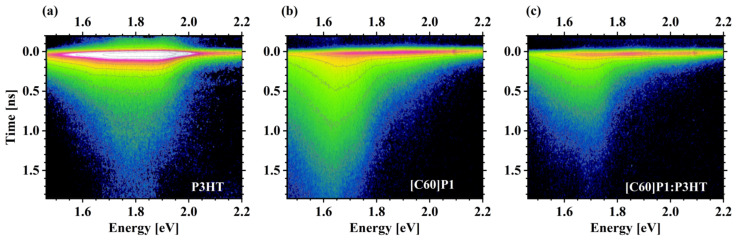
TRPL spectra of (**a**) P3HT, (**b**) [C60]P1, and (**c**) blend of [C60]P1 and P3HT at 3:2 molar ratio. Apparently, the luminescence of the blend is weaker and decays faster than those of the components. The plotted data are contour maps of luminescence intensity as a function of time (vertical scale) and energy (horizontal scale). The intensity is plotted in a logarithmic scale, so the intensity from contour to contour grows *e* times (where *e* is the Euler number).

**Figure 8 materials-15-06908-f008:**
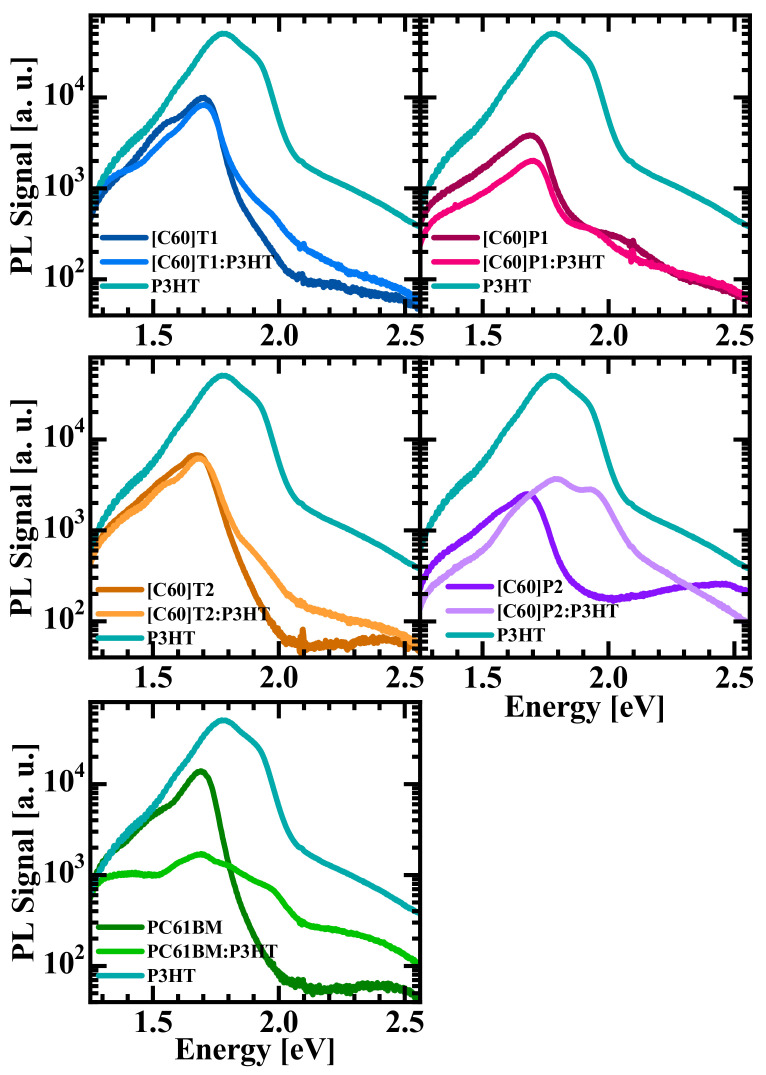
PL spectra of P3HT, fullerene derivative, and their blend.

**Figure 9 materials-15-06908-f009:**
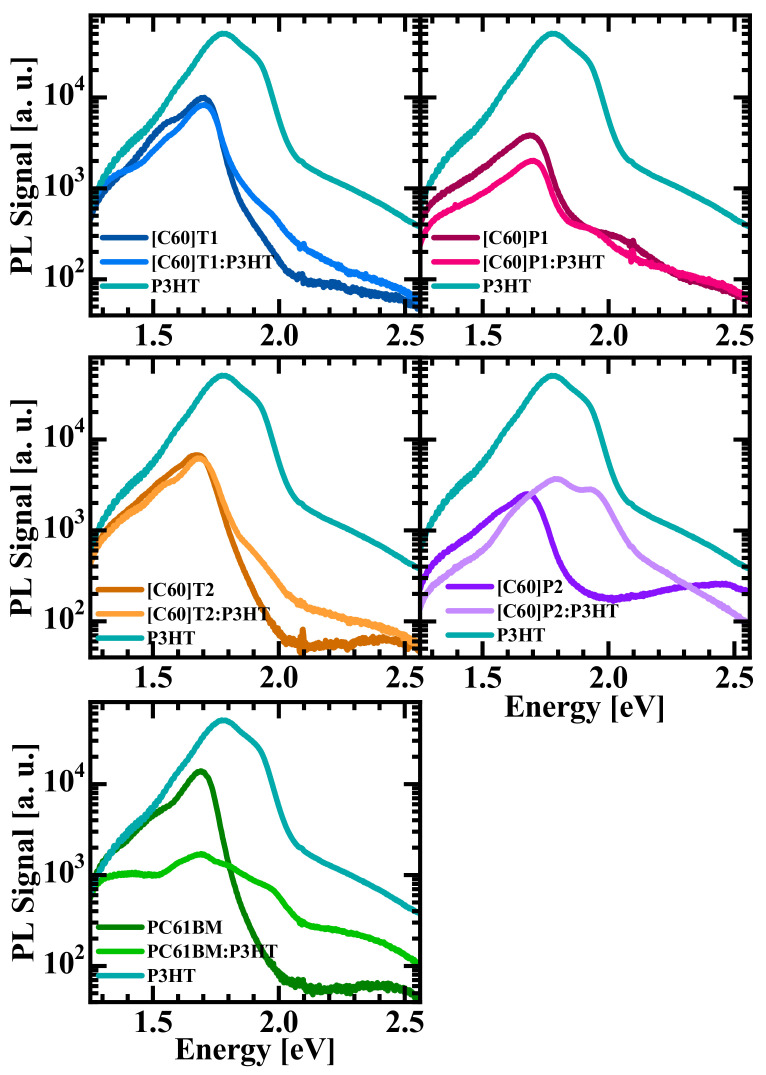
PL signal of the studied fullerene derivatives as integrated in the energy range of 1.54–1.8 eV for pure fullerenes and for fullerene–P3HT blends. The fits with single exponential decays are depicted with dashed lines.

**Table 1 materials-15-06908-t001:** HOMO (also HOMO-1 and HOMO-2 for selected compounds) and LUMO energies of the optimized fullerene derivatives obtained by DFT calculations. Additionally, spatial models of corresponding HOMO and LUMO with yellow and purple lobes corresponding to positive, and green and blue lobes to negative isosurface values.

Material	PC61BM	[C60]T1	[C60]P1	
Spatial distribution of HOMO/HOMO-1 and LUMO isosurfaces	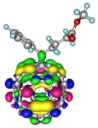	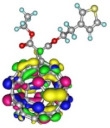	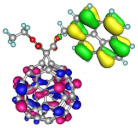	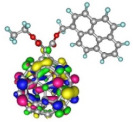 HOMO-1
LUMO [eV]	−3.09	−3.12	−3.10	
HOMO [eV]	−5.66	−5.70	−5.50	−5.68
Frontier Molecular Orbital gaps [eV]	2.57	2.58	2.40	2.58
Material	[C60]T2	[C60]P2		
Spatial distribution of HOMO/HOMO-1/HOMO-2 and LUMO isosurfaces	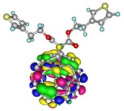	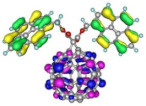	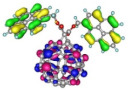 HOMO-1	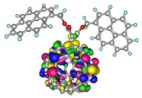 HOMO-2
LUMO [eV]	−3.15	−3.06		
HOMO [eV]	−5.73	−5.48	−5.50	−5.65
Frontier Molecular Orbital gaps [eV]	2.58	2.38	2.44	2.59

**Table 2 materials-15-06908-t002:** g-tensor components obtained for all studied fullerene-derivative–P3HT blends.

	*g_x_*	*g_y_*	*g_z_*	*g_iso_*
PC61BM^−^	2.0001	2.0001	1.9992	1.9998
P3HT^+^	2.0026	2.0023	2.0012	2.0020
[C60]T1^−^	2.0005	2.0005	1.9998	2.0003
P3HT^+^	2.0032	2.0022	2.0012	2.0022
[C60]P1^−^	2.0003	2.0001	1.9998	2.0000
P3HT^+^	2.0031	2.0022	2.0011	2.0021
[C60]T2^−^	1.9999	1.9999	1.9999	1.9999
P3HT^+^	2.0034	2.0022	2.0011	2.0022
[C60]P2^−^	2.0002	1.9999	1.9999	2.0000
P3HT^+^	2.0034	2.0022	2.0011	2.0022

## Data Availability

The data presented in this study are available on request from the corresponding author.

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
