# Peer review of "Optical Properties and Light-Induced Charge Transfer in Selected Aromatic C60 Fullerene Derivatives and in Their Bulk Heterojunctions with Poly(3-Hexylthiophene)"

_materials, 2022, doi:10.3390/ma15196908_

Round 1

Reviewer 1 Report

This article systematically studied C60 and a series of its derivatives. Both optical and electronic structures are carefully investigated. However, there are several questions regarding the TRPL measurement and DFT calculations.

1.      Is TRPL measured in solution or on thin film? It would be clear to mention this in the experimental section.

2.      Some TRPL spectra exhibited a double exponential decay character. It would be helpful to fit the curves to a double exponential decay function. The comparison of short-lifetime terms would be useful to clarify the recombination pathway.

3.      I understand that the DFT calculations for large systems are often limited by computational resources. Therefore, the double-zeta 6-31G(d) basis set is the minimum acceptable basis set for geometry optimization. However, the double-zeta basis set is not appropriate for the following electronic structure calculations. I highly suggest that the authors do one more single-point calculation using triple-zeta basis sets, which could remarkably improve the quality of the MO energy levels.

4.      Table 1. The header “HOMO orbital” is inappropriate. HOMO is the highest occupied molecular orbital. The “orbital” after HOMO is redundant. I understand that the author means the iso-surfaces of HOMO and LUMO. It is the spatial distribution or say localization of HOMO and LUMO.

5.      Table 1. The author also included the gap between HOMO-1 and LUMO. But the header is LUMO-HOMO. The header can be addressed as “FMO gaps” or “LUMO – HOMO / HOMO-1”.

6.      Line 128. No imaginary frequency does not guarantee a global minimum. A local minimum sometimes also yields non-imaginary frequency. I suggest the author delete this expression.

7.      In future works, I strongly suggest the author add D3(BJ) semi-empirical dispersion correction. This could significantly enhance the accuracy of geometry optimization without extra cost of computational resources for DFT calculations.

8.      The author actually performed TDDFT calculations. This should be mentioned in the experimental section.

9.      How is the LUMO looks like for these compounds? Why they are not shown in Table 1?

10.   How is the excitation states contributed by certain MO pairs? For example, C60 showed two states with notably large oscillator strength. How are these states contributed by their MO pairs? Are they dominated by a certain MO pair? If not, is NTO analysis applicable?

11.   I suspect that some excited are charge transfer (CT) states since the HOMO and LUMO are well separated. Is B3LYP appropriate for CT excitation? Or the states with high oscillator strength has a local excitation (LE) character? Related post-analysis and discussions are missing here.

Author Response

Dear Reviewer,

Thank you for a very insightful reading and reviewing the manuscript. Below you can find a exhaustive explanation of each of your remarks.

  1. Is TRPL measured in solution or on thin film? It would be clear to mention this in the experimental section.

It was measured on thin film. The following sentence ending in section 2.1 was added to satisfy this remark:

However, in these cases, small droplets of the solution were applied onto a glass substrate (UV-Vis) or copper foil (PL and TRPL) and let dry at 60 °C to create a thin film.

  1. Some TRPL spectra exhibited a double exponential decay character. It would be helpful to fit the curves to a double exponential decay function. The comparison of short-lifetime terms would be useful to clarify the recombination pathway.

The curves were added to the Figure 9. The fitting using double exponential decay was not possible since it creates significant fitting errors. We used 2 independent single exponential decay functions to fit the respective range of the measurements separately. The table with the exact times was added to supplementary materials (Table S4). Due to slightly different fitting bounds in the short time region (0 – 0.25 ns or 0.3 ns depending on the signal curvature), the obtained times are slightly different, yet the general conclusion regarding these results is the same, i.e. the most efficient hole transfer takes place in PC61BM sample.

.      I understand that the DFT calculations for large systems are often limited by computational resources. Therefore, the double-zeta 6-31G(d) basis set is the minimum acceptable basis set for geometry optimization. However, the double-zeta basis set is not appropriate for the following electronic structure calculations. I highly suggest that the authors do one more single-point calculation using triple-zeta basis sets, which could remarkably improve the quality of the MO energy levels.

Thank you very much for this comment. It is true that using a larger basis set usually produces more accurate results, usually better reflecting the results of experimental measurements. As suggested by Reviewer 1, we performed additional sp calculations (optimized geometry at the B3LYP/6-31G (d) level) using the same functional as used in the research (B3LYP) and increased basis set (triple-zeta basis sets), i.e. 6-311G(d, p). The energy values obtained in this way for HOMO and LUMO are given in Table S5.

As suggested, the following paragraph has been added to the manuscript (section 3.2):

Due to the size of the studied molecules, to optimize their geometry and calculate the corresponding energies, we used a relatively small double-zeta basis sets 6-31G (d). To compare whether calculation using larger triple-zeta basis sets can qualitatively change conclusions resulting from HOMO and LUMO energy estimates, single-point calculations for the optimized geometries were performed using triple-zeta basis sets , i.e. 6-311G (d, p). The energy values obtained in this way for HOMO and LUMO are included in Table S4. It can be seen that in general there is no change in the sequence of levels for all fullerene derivatives, and the energies of the levels are shifted downwards by about the same, -0.4eV, energy. Therefore the following discussion and conclusions remain valid also in the larges basis.

  1. Table 1. The header “HOMO orbital” is inappropriate. HOMO is the highest occupied molecular orbital. The “orbital” after HOMO is redundant. I understand that the author means the iso-surfaces of HOMO and LUMO. It is the spatial distribution or say localization of HOMO and LUMO.

We have changed the above-mentioned headers in Table 1 to Spatial distribution of HOMO/HOMO-1/HOMO-2 and LUMO isosurfaces.

  1. Table 1. The author also included the gap between HOMO-1 and LUMO. But the header is LUMO-HOMO. The header can be addressed as “FMO gaps” or “LUMO – HOMO / HOMO-1”.

We have changed the above-mentioned headers in the Table 1 to Frontier  Molecular Orbital gaps.

  1. Line 128. No imaginary frequency does not guarantee a global minimum. A local minimum sometimes also yields non-imaginary frequency. I suggest the author delete this expression.

As suggested, statements „This supposition was proved by vibrational frequency calculations which turned out to give all positive energy values when deviating from the designated configuration. Thus, the optimized geometries reflected reasonable structures of the global minimum.” in section 2.2 was deleted.

  1. In future works, I strongly suggest the author add D3(BJ) semi-empirical dispersion correction. This could significantly enhance the accuracy of geometry optimization without extra cost of computational resources for DFT calculations.

Thank you for this comment.

  1. The author actually performed TDDFT calculations. This should be mentioned in the experimental section.

We have added the following statement in section 2.2 to satisfy this remark:

Density Functional Theory (DFT) and Time-Dependent Density Functional Theory (TDDFT).

Also, we have added the following sentence in the same section:

UV-VIS spectra has been calculated by TDDFT as implemented in Gaussian package. Computations were performed with the use of B3LYP functional, in the base 6-31 G(d). Number of excited states was 108.

  1. How is the LUMO looks like for these compounds? Why they are not shown in Table 1?

We have added also the LUMO spatial distribution in Table 1.

  1. How is the excitation states contributed by certain MO pairs? For example, C60 showed two states with notably large oscillator strength. How are these states contributed by their MO pairs? Are they dominated by a certain MO pair? If not, is NTO analysis applicable?

Thank you for raising this question. This is an interesting point, however insightful discussion on this point might dilute the whole paper even more, straying away further from its main research topic. Moreover, the editor politely asked us to be more concise and focused on the most relevant results to considerably shorten the paper.

  1. I suspect that some excited are charge transfer (CT) states since the HOMO and LUMO are well separated. Is B3LYP appropriate for CT excitation? Or the states with high oscillator strength has a local excitation (LE) character? Related post-analysis and discussions are missing here.

In fact, plotted results of calculations suggest that charge transfer take place in an excited state. Perhaps, it could be described more precisely with the use of other tools such as CAM-B3LYP functional, along with the use of a larger basis. However,  in our opinion, its influence could not be very substantial. The scale of charge transfer should be larger for complexes of donor and acceptor, as studied for instance in:

  1. Korona and D. Rutkowska-Żbik: A theoretical study of elementary building blocks for organic solar cells: influence of perturber molecule on electronic spectrum of PCBM. Comput. Theor. Chem. 1040-1041 (2014) 243-258

where the CAM-B3LYP functional has been used. More detailed calculations including comparison of various functionals and bases deserve separate, in-depth studies. Within this paper, we wanted to gain some basic information concerning HOMO-LUMO levels and optical spectra.

We hope that will be satisfied with the explanation and revisions we have made throughout the manuscript.

Best regards,

Authors

Reviewer 2 Report

Friday the 02 of September 2022 

Review on the manuscript  "Optical properties and light-induced charge transfer in selected aromatic C60 fullerene derivatives and in their bulk heterojunctions with poly(3-hexylthiophene) " submitted to Materials.

This work is an experimental study comparing four C60 fullerene derivatives. The authors characterize the optical properties and the efficiency of the charge transfer process for the materials alone and associated with P3HT to form a heterojunction. The characterizations include time-resolved photoluminescence (TRPL), UV-vis optical absorption and electron spin resonance (ESR). The authors conclude that the absorption processes of fullerene derivatives are little modified by grafting (not surprising). The ESR and TRPL experiments suggest that the asymmetric derivatives transfer the generated photo charges better than the more symmetric ones. The optical absorption experiments were modeled by DFT. 

This work is overall relatively good, but the goal of authors of a paper should be to be read, understood and cited. In the current state of this manuscript, I would not quoe it. It is far too long for the amount of information it contains. Many things, both experimentally and computationally, are repetitions, either internally in the article or from other publications. I recommend to limit the dilution as much as possible and to write only new, original and relevant things. The pleasure of reading will only be reinforced. 

-----------------

Beware of typos. 

How was the normalization of the absorbance done? 

The graphs are of relatively good quality, the color codes are respected. 

   Remember to add the color code in Figure 2.

   1E-4 (figure 6). 

Author Response

Dear Reviewer,

Thank you for your comments. To satisfy your remarks, we considerably shorten the article. Especially, the parts where some repetitions, citing other works and non-relevant information are now removed or rewritten in more concise way. We deleted or revised many paragraphs throughout the whole manuscript, too many to exhaustively describe them here. The uploaded word document has Track Changes set on, so you can easily investigate every single sentence removed or revised using this tool.

  1. Beware of typos.

Thank you for this comment. We tried our best to remove unintentional typographical errors in the text and generally polish the language.

  1. How was the normalization of the absorbance done? 

The obtained spectra was normalized to the intensity of 3.7 eV peak. The following sentence was added in 3.1 section:

Figure 3a shows the absorbance normalized to the intensity of the 3.7 eV peak absorbance of pure fullerene derivatives (including PC61BM) at room temperature in the energy range of 1.75 to 4.0 eV.

  1. The graphs are of relatively good quality, the color codes are respected.

Thank you for this comment and appreciation of our efforts.

  1. Remember to add the color code in Figure 2.

The labels below the molecules pictures are now color matched with the lines presented across the entire paper.

  1. 1E-4 (figure 6)

We assume that you mean the tick label should be 0.0001 instead of 1E-4. We corrected this tick label to match the rest of the labels.

We hope that this explanation and revisions done in the manuscript make a sufficient response for your remarks.

Best regards,

Authors

Round 2

Reviewer 1 Report

Thanks for the detailed response. All questions are addressed properly. The paper is now ready to be published.